# Germination and the Initial Seedling Growth of Lettuce, Celeriac and Wheat Cultivars after Micronutrient and a Biological Application Pre-Sowing Seed Treatment

**DOI:** 10.3390/plants10091913

**Published:** 2021-09-14

**Authors:** Dobrivoj Poštić, Ratibor Štrbanović, Marijenka Tabaković, Tatjana Popović, Ana Ćirić, Nevena Banjac, Nenad Trkulja, Rade Stanisavljević

**Affiliations:** 1Institute for Plant Protection and Environment, 11040 Belgrade, Serbia; pdobrivoj@yahoo.com (D.P.); ratibor.strbanovic@yahoo.com (R.Š.); tanjaizbis@gmail.com (T.P.); trkulja_nenad@yahoo.com (N.T.); 2Maize Research Institute Zemun Polje, 11185 Belgrade, Serbia; mtabakovic@mrizp.rs; 3Institute for Biological Research Siniša Stanković—National Institute of the Republic of Serbia, University of Belgrade, 11000 Belgrade, Serbia; ancic@ibiss.bg.ac.rs (A.Ć.); mitic.nevena@ibiss.bg.ac.rs (N.B.)

**Keywords:** *Bacillus* spp., boron, Coveron^®^ seed and seedling quality, *Trichoderma* spp., zinc

## Abstract

Seed treatments with zinc, boron, biostimulant Coveron and MIX (zinc + boron + Coveron) were applied to three lettuce and three celeriac cultivars. Seeds of three wheat cultivars were treated under laboratory conditions with *Trichoderma harzianum* and eight *Bacillus* spp. Seed germination, seedling growth, and the presence of the following pathogens were determined: *Fusarium* sp., *Alternaria* sp., *Penicillium* sp., and *Mucor* sp. The Coveron treatment was the most effective on lettuce seeds tested in the germination cabinet. Seed germination was higher by 4% than in the control. Alternatively, germination of seeds treated with boron in the greenhouse was higher by 12% than in the control. The Coveron treatment had the highest effect on the shoot length, which was greater by 0.7 and 2.1 cm in the germination cabinet and the greenhouse, respectively. This treatment was also the most effective on the root length. Zn, B, and MIX treatments increased celeriac seed germination by 14% in the germination cabinet. The Zn treatment was the most efficient on seeds tested in the greenhouse. The germination was higher by 15%. A significant cultivar × treatment interaction was determined in both observed species under both conditions. The maximum effect on wheat seed germination (8%) was achieved with the *T. harzianum* treatment in the Salazar cultivar. A significant interdependence (*p* ≤ 0.01 to *p* ≤ 0.001) was established between seed germination and the seedling growth. The interrelationship between seed germination and pathogens of all cultivars was negative.

## 1. Introduction

Lettuce (*Lactuca sativa* L.) is a plant species of the family Asteraceae. World production of lettuce is about 21 million tonnes per year, mainly under temperate climatic conditions [1]. It is traditionally used fresh. Its consumption has a beneficial effect on human health [2]. Celeriac *(Apium graveolens* L.) is a plant species of the family Apiaceae. In human nutrition, the leaf stalks, leaves, and roots are used as the ingredient in soups, salads, and various other side dishes. It has diuretic, antiseptic, anti-allergy, and anti-inflammatory effects [3]. It also has medicinal properties for the treatment of cardiovascular diseases, high blood pressure, and liver diseases in humans [4].

Both species can be grown in the open field or in any type of a sheltered place (plastic-house, greenhouse) in the organic or the conventional cultivation system. Regardless of the cultivation system, it is necessary that seed germination and the initial seedling growth are high. De-germination can occur due to many reasons: inadequate temperature, light, sowing depth. However, it can be a consequence of the use of poor quality seeds. Growers of both species often complain about low seed germination and a weak initial growth of seedlings.

Zinc is involved in many biochemical and physiological processes of germination and/or photosynthesis: protein synthesis, membrane building, cell elongation, resistance to abiotic stress, metabolism of carbohydrates, lipids, nucleic acid synthesis, stimulation of auxin synthesis [5,6].

Boron affects the metabolic processes of nucleic acids, carbohydrates, proteins, indol acetic acid. It affects the cell wall synthesis, metabolic activation of phenol, calcium use, cell division [5,7].

Fungal species *Trichoderma* have a pronounced bio-stimulant effect in germination and other physiological processes, hence in the past decade, it has been widely used in agriculture [8]. *Glomus* spp. affects the increase of the shoot and root weight [9]. Wheat (*Triticum aestivum* L.) on a global scale, in addition to rice, is a cereal that is mostly used in human nutrition. Seed-borne fungi can affect seed quality losses such as abortion, rot, necrosis, reduction or elimination of germination capacity, then seedling damage and their nutritive value and also variation in plant morphology [10]. Among the most common plant pathogens in association with wheat seed worldwide are fungi from genera *Alternaria* sp., *Aspergillus* sp., *Curvularia* sp., *Drechslera* sp., *Fusarium* sp., *Penicillium* sp., *Mucor* sp., *Pythium* sp., *Rhizopus* sp. [11]. Yield losses caused by seed-borne pathogens are found to be among 15 to 90% of untreated wheat seeds grown in fields [12]. Regarding that, numerous research have been conducted for the possibility of biological control of undesirable seed pathogens and increase of germination [13,14]. Moreover, there are various investigations in order to increase the areas in organic farming [15]. Variation of the total content of polyphenols and phenolic acids in einkorn, emmer, spelt, and common wheat grain is a function of genotype, wheat species, and crop year [16]. One of the currently used strategies to preserve the health of cultivated plants in organic agriculture is the use of microorganisms that manifest beneficial effects on plant growth and development [17]. Since fungi *Trichoderma* spp. displayed broad-spectrum of antagonistic activities against various plant pathogens [18,19] it has been considered as the major biocontrol agent in biopesticide industry [20] and used for seed treatment [21]. In recent years, *Trichoderma* has acquired high importance because of its fungicidal and fertilizing potential [22,23]. *Bacillus* species also have a positive effect on plant growth and development by regulating the metabolism of plant hormones (auxins, cytokinins, gibberellins), which affects nitrogen fixation and phosphate availability and thus affects seedling growth. In addition, *Bacillus* spp. produce substances such are antibiotics, enzymes, hydrogen cyanide, which are an important tools in the biological control of plant pathogens [24].

According to Glick, [25] *Bacillus* spp. belong to plant growth-promoting bacteria (PGPR) and can increase the plant growth through direct or indirect mechanisms. Direct mechanisms involve biological processes such as biological nitrogen-fixation, solubilization of complex organic or inorganic nutrients, sequestering of iron via siderophore production, and modulating phytohormone levels such as gibberellins, cytokinins, indole acetic acid (IAA), and ethylene.

The aim of this study is: (i) to determine germination and the initial growth of seedlings of three lettuce and three celeriac cultivars after pre-sowing seed treatments with zinc, boron, then mixture of zinc, boron, and the biological product—Coveron, and Coveron alone, in the germination cabinet (seeds placed on sand) and the greenhouse conditions (seeds placed on soil—these conditions are close to conventional production conditions); (ii) to examine three cultivars of wheat seed for the effect of treatments with *Trichoderma harzianum* isolate and eight isolates of *Bacillus* spp. on the germination and seedling development and influence on the presence of fungal seed plant pathogens under laboratory conditions.

## 2. Results and Discussion

Extremely high temperatures before the seed reaches physiological maturity to reduce the plant ability to form assimilates are necessary for the synthesis of storage compounds required for germination [26]. Moreover, due to high temperatures the seed suffers physiological damage to the extent that its ability to germinate is reduced [27]. However, when the temperature is similar and the application of cropping practices in production is the same, then the conditions for the growth, development, and maturation are the same, and the quality of seeds is uniform [28]. In our tests, the influence of location was not significant in any of the examined species, probably because the distance of production plots was not more than two kilometers. Seed production was performed in the vicinity of the city of Subotica for lettuce and celeriac, and near the city of Pančevo for wheat. During the filling period and grain maturity, the weather conditions in the two-year testing period were similar on plots for lettuce, celeriac, and wheat production, which probably affected the year without affecting seed germination and seedling quality.

Effects of the cultivar × location (A × C), cultivar × year (A × D), treatment × location (B × C), treatment × year (B × D), and location×year (C × D) interactions on germination and the seedling growth were not significant in any of the species (*p* ≥ 0.05). On the other hand, effects of factors A and B, as well as their interactions, on the observed traits in the germination cabinet and the greenhouse were significant in both species (*p* ≤ 0.05 to *p* ≤ 0.01).

### 2.1. Lettuce Seed

#### 2.1.1. Germination

Seed priming is an extensively used practice to increase the rate and uniformity of seed germination that can be a suitable alternative for wheat seedlings to provide the higher yield. The seed treatment using biological agents including a number of *Trichoderma* species was an eco-friendly method used to improve germination in other important grasses such as rice [29], maize, and oaths [30,31,32,33]. Although *Trichoderma* species showed the wide interaction potential associating with numerous plant species, some specificity in *Trichoderma* spp. plant species/cultivar relation has been also found [34].

Zinc is involved in the stabilization of cell membranes, detoxification of free radicals, secondary metabolism, and other biochemical and physiological processes during germination and the early growth [35]. Boron affects metabolism of starch, thus significantly affects the rate of both, germination and the initial seedling growth [36]. According to Mondal and Bose [37], micronutrients can be applied to crops through soil, leaves, or seeds. The micronutrient application through the seed treatment has numerous advantages: it is easy to apply; it is cost effective; and it highly affects germination and the initial seedling growth. As stated by Aboutalebian et al. [38], the wheat seed priming with zinc resulted in the significant increase in germination and the initial seedling growth. Furthermore, maize seed priming with zinc led to the increase of germination under conditions of both, laboratory and filed [39,40], which resulted in the yield increase by up to 27% [41]. According to Rahman et al. [42], rice seeds treated with boron had higher and faster emergence of the initial seedling growth, but it was important that the boron concentration was 0.001% or 0.1%, while boron concentration of 0.5% was ineffective. The application of the optimal Zn concentration to chili seeds affected the increase of germination by 4%, root length by 0.45 cm, and the shoot length by 0.09 cm [43]. Poštić et al. [44] applied the combination of *Trichoderma* + *Glomus* to pepper seeds and determined the increase of seed germination by 4–6% and 3–4% during the first and the second year of investigation, respectively.

Similar results were reported by Štrbanović et al. [45]. These authors treated pepper seeds with zinc and boron prior to sowing and found the germination rate to be higher by 5% and 6%, shoot growth higher by 1.3 cm and 1.2 cm, and the root growth higher by 0.8 cm. Nevertheless, there are studies in which seeds were treated with zinc or boron and no significant increases in germination and/or the seedling growth were achieved: for instance, rice [46] and wheat [47]. Therefore, based on literature data, the effect of the boron and zinc application on germination and initial seedling growth significantly differs based on plant species, genetics, i.e., a cultivar or a hybrid of the same species, the application method, or the concentration of microelements.

Lettuce seeds were stored in the germination cabinet, and after seven days, germination was higher in treatments than in the control: by 4% (Coveron), 3% (Zn and MIX), and by 2% (B) (the average of all cultivars). However, each cultivar differently responded to treatments: Coveron was the optimal treatment for cultivars Genius and Majska kraljica, with germination higher by 4% and 5%, respectively; Zn and B were the optimal treatments for cultivar Endivia and they increased germination by 5% (Table 1b).

After 10 days of storage of treated seeds in the greenhouse, germination compared to the control, was higher by 12% (B), 11% (Zn and MIX), and 10% (Coveron) (Table 1b). MIX was the optimal treatment for cultivar Genius and it increased germination by 16%; Zn was the optimal treatment for cultivar Majska kraljica and it also increased germination by 16%, while B was the optimal treatment for cultivar Endivia and it increased germination by 7%. The resulting differences in seed germination were statistically significant (*p* ≤ 0.05) in all observed lettuce cultivars in both, the germination cabinet and the greenhouse (Table 1b).

#### 2.1.2. Shoot Length

Generally, in comparison to the control, the highest effect, on average, in the germination cabinet, on the shoot length was expressed by the Coveron treatment and the length was greater by 0.5 cm and 1.2 cm in cultivars Genius and Endivia, respectively. Cultivar Majska kraljica was mostly affected by B and the shoot was longer by 0.6 cm (Table 2a). The highest effect, on average, on all cultivars in the greenhouse, was recorded with Coveron, and in comparison to the control, the shoot length was longer by 2.1 cm. The effect of all treatments on shoot length was the same, with a coefficient of variation (CV) of no more than 4.5%. The effect of the cultivar × Coveron treatment interaction in the greenhouse was the highest on the Endevia seeds (8.1 cm). The seed treatment with Coveron was optimal for all observed cultivars, while the optimal Zn or B treatment differed over cultivars (Table 2b).

#### 2.1.3. Root Length

The average root length of seeds tested in the germination cabinet, in comparison to the control, was longer by 0.5 cm, 0.4 cm, 0.3 cm, and 0.3 cm when seeds were treated with Coveron, MIX, Zn, and B, respectively (Table 3a). MIX was the optimal treatment for cultivar Genius with the root length increased by 0.4 cm. Coveron was the optimal treatment for cultivar Majska kraljica, and the root length increased by 0.6 cm, while Coveron and Zn were the most efficient treatments for cultivar Endivia with the root length longer by 0.8 cm and 0.7 cm, respectively.

In the greenhouse, the increase in the root length was 0.3 cm and 0.2 cm when Coveron and remaining treatments, respectively, were applied. The response of cultivar Genius to Zn and Coveron was the best and the increase in the root length was 0.3 cm. Coveron was the most efficient for cultivar Majska kraljica (root longer by 0.3 cm), while Zn and B treatments were the most efficient for cultivar Endivia (root was also longer by 0.3 cm) (Table 3b).

According to the F test, the effect of factors, location (C), and year (D) was not significant (further on in the text the average is presented). The trend of improved germination and the initial growth of seedlings was observed in lettuce seeds of all cultivars in all treatments in both investigation variants (the germination cabinet and the greenhouse), but the difference was not significant (*p* ≥ 0.05) between the control and the Zn and B treatments in cultivar Genius and cultivar Majska kraljica, respectively, as well as between the control and the Coveron treatment in cultivar Endivia. The difference in the root growth between the control and Zn and B treatments in cultivar Genius was not significant. Moreover, the difference in the root growth between the control and the MIX treatment was not significant in the cultivar Endivia (Table 4).

The PCA was applied to the lettuce seeds in order to evaluate the relation between cultivars and applied methods. The first and the second principal components (PC-1 and PC-2) accounted for 56.04% and 19.14%, respectively, of the total data variance, i.e., their mutual projections (Figure 1). This analysis also confirmed that the treatments and cultivars were correlated and classified into the same group, A1 and A2, except for B—Genus (GE), while on the other hand, control treatments were in the second group (B1 and B2) (Figure 1).

### 2.2. Celeriac Seed

#### 2.2.1. Germination

Germination of three celeriac cultivars was uneven in seeds tested in the germination cabinet. The greatest variation in germination (CV 12.1%) was recorded in the untreated seeds (control), while the Zn treatment equally affected all cultivars with the average germination of 88% and CV 3.64%. The most variable germination within the treatment + control was determined in the celeriac cultivar Cazar (CV 11.0%). The highest increase in germination in comparison to the control was recorded in cultivar Cezar when Zn was applied (23%). The B treatment resulted in germination higher by 10% in cultivar Omega, while effects of B, MIX, and Coveron treatments were equally good for cultivar Praški krupni (germination was higher by 15%) (Table 5a).

The Zn treatment showed the highest efficiency on germination of seeds tested in the greenhouse; the germination was higher by 15% than the control. The significance of the cultivar × treatment interaction was the highest in cultivar Cezar with the variation between the applied treatments and the control of 12.7%. Zn was the optimal treatment for cultivar Cezar, while the B treatment was the optimal treatment for cultivars Omega and Praški krupni (Table 5b).

#### 2.2.2. Shoot Length

After measuring shoots in seedlings obtained in the germination cabinet, the highest (0.7 cm), i.e., the lowest (0.5 cm) increase in the shoot length in relation to the control was recorded in seeds treated with Coveron, i.e., with Zn and B, respectively (average of all observed cultivars).

The treatment effects differed over cultivars. The optimal treatments for cultivar Cezar were B and MIX (shoot growth higher by 0.3 cm in relation to the control), while Coveron was the optimal treatment for cultivars Omega and Praški krupni (growth higher by 0.8 cm and 1.0 cm, respectively in relation to the control) (Table 6a).

The treatment MIX showed the highest effect on the shoot growth under the greenhouse conditions (CV 25.1%). The significance of cultivar×treatment interaction for the shoot growth was the most pronounced in the cultivar Cezar (CV 22.8%) (Table 6b).

#### 2.2.3. Root Length

The root length of seedlings in the germination cabinet varied from CV 4.22% (Zn treatment) to CV 9.17% (MIX treatment). The differences in the root length over cultivars and treatments varied from CV 10.5% to 23.0%. The longest root of 1.8 cm was obtained by the application of the Coveron, i.e., MIX treatment on seeds of cultivars Omega, i.e., Praški krupni, respectively (Table 7a).

On the other hand, the root length of seedlings in the greenhouse ranged from 2.7 cm (control) to 4.2 cm (Coveron treatment). The greatest variation in the expression of this trait was recorded when Zn was applied (CV8.33%), and the highest effect of the cultivar on the root length variation was observed in cultivar Cezar (Table 7b).

The difference between the control and the Zn treatment was not statistically significant (*p* ≥ 0.05) for the root length for cultivars Omega and Cezar, and between the control and the MIX treatment for cultivar Omega in the greenhouse (Table 8).

The PCA was applied to the celeriac seeds and the first and the second principal components (PC-1 and PC-2) accounted for 52.48% and 35.49%, respectively, of the total data variance, i.e., their mutual projections (Figure 2). This analysis also confirmed that the treatments and cultivars were correlated and classified into the same group, A1 and A2, except for the treatments in cultivar Cezar (CE), while on the other hand, control treatments were in the second group (B1 and B2) (Figure 2).

### 2.3. Wheat Seed

#### 2.3.1. Effects of Factors: Years (Y)/Cultivars (C)/*Trichoderma harzianum*, *Bacillus* spp. and Treatment (T)

Effects of the year, and the Y x C interaction and Y x T interaction were not significant (*p* ≥ 0.05), values presented are the two-year averages. Effects of the factors cultivar and treatments, as well as their interaction on all examined traits were significant (*p* ≤ 0.05 to *p* ≤ 0.01) (Table 9).

#### 2.3.2. Germination

In general, in all the examined cultivars, *T. harzianum* seed treatments showed the highest efficiency on seed germination, which was statistically significantly higher compared to the control (Table 10). Using *T. harzianum* the improvement of germination of wheat seeds ranged from 4% to 8% and mostly depended on the wheat cultivar. In addition to the treatment with *T. harzianum*, similar efficiency in increasing germination but with a slightly lower level of significance (*p* ≤ 0.05) was shown by the *Bacillus* isolate B5 in all three wheat cultivars, as well as isolates B7 and B8 in Salazar and Zemunska rosa, respectively (Table 10). On the other hand, between the treatment with *T. harzianum* and *Bacillus* isolates B1, B2, and B4, a significant difference was found in germination in the cultivar Apicol (*p* ≤ 0.05).

In the cultivar Salazar, a significant difference was found between the effects of *T. harzianum* and *Bacillus* isolates B2, B4, and B6. In the cultivar Zemunska rosa, a significant difference was found between *T. harzianum* and *Bacillus* isolates B1, B3, B4, and B6. The examined treatments had an increase in germination by 4%, compared to the control in the cultivars Apicol (treatments *T. harzianum* and B5) and Zemunska rosa (treatments *T. harzianum* and B7). In comparison to the control, the applied treatments affected the increase in germination by 4% in the cultivars Apicol (treatments *T. harzianum* and B5) and Zemunska rosa (treatments *T. harzianum* and B7). The largest increase of 8% was found in cultivar Salazar after the treatment with *T. harzianum* (Table 10).

Therefore, the differences in germination promoting effects could be attributed to the specific association released between individual wheat cultivar and particular *Trichoderma* strain. Similarly, four various *Trichoderma* strains applied in wheat BRS 264 displayed different promoting potential ranging from 3% to 8%, while *T. harzianum* ESALQ 1306 provided the best results regarding germination and seedling performances under greenhouse conditions [48]. *Trichoderma* may improve germination by increasing the synthesis of endogenous plant growth regulators by up-regulation of genes for their synthesis and by down-regulation of genes included in their catabolism [49]. Since gibberellic acid (GA3) can improve seed germination by promotion of activity on hydrolytic and proteolytic enzymes that act to mobilize the food reserves from the cotyledons or endosperm, at least one of the promotive effect of *T. harzianum* IS005-12 on wheat seeds germination is attributed to increasing of GA3 level. In cherry rootstocks *T. harzianum* strain T22 increased the content of GA3 in leaves by 71% and in roots even by 143% [50]. Harman et al. [51] reported that treatment of soybean seeds by conidia of *T. harzianum* T22 enhanced phase III imbibitions (cell elongation, followed by radicle protrusion). Roots of these seeds were larger and more robust as evidenced in soybean plants in which an increase of 123% in yield was obtained when inoculated with T-22 [51]. Furthermore, Hoyos-Carvajal et al. [52] reported beneficial effects on germination and seedling emergence, since the solubilized nutrients become more available for root uptake [52].

Species of the genus *Trichoderma* have developed several mechanisms that con-tributed to increasing plant resistance to diseases, growth and development, and root morphology, as well as seed germination [53,54]. The plant growth promotion by *Trichoderma* has been observed in many crops. The present study shows the results of seed treatments of three wheat cultivars with *T. harzianum*. The results indicated a positive effect of this saprophytic fungus on seed germination and the seedling growth. According to Gravel et al. [55], *Trichoderma* spp. produced indole-3-acetic acid (IAA), which physiologically affected the elongation and the growth of cells, and as a result stimulated the accelerated growth of tomato seedlings. The *T. harzianum*-based biological preparation (Ecotrich WP) showed similar effects on the development of soya bean roots [56], which is consistent with and explains our results on seed germination and the initial seedling growth of lettuce and celeriac (Table 1 and Table 2). *Trichoderma* spp. when absorbed by seeds of wheat and Italian ryegrass promoted germination, the initial seedling growth, resistance of newly emerged seedlings to diseases caused by pathogens [54,57]. However, mechanisms of action of *Trichoderma* strains are influenced by genetic variability of *Trichoderma*, genetic variability of the plant species whose seeds are treated as well as their interactions. Therefore, whether effects will be positive or negative are unpredictable [14,54].

Wheat seed study, the impact of the year was not significant (Table 9) on properties of wheat seeds, this is probably because both years were similar in terms of climatic conditions in which production took place (not shown). Studies of wheat seeds show that the impact of the year was not significant (Table 9) on the wheat seed properties. This is probably because both years were similar with regard to weather conditions under which production was performed. It is recommended that the wheat production in the world is carried out with the reduced use of pesticides or without their use [15,53,58,59]. These measures reduce the risk of negative effects of wheat products on human health [24,58]. Bacteria that produce extracellular antifungal compounds that indirectly stimulate the self-defense system of the host plant are capable of acting antagonistically on pathogens. Because of this action, they are ideal for biological control [60]. According to Ji et al. [15], organic cereal production methods result in significantly lower pesticide residues in wheat flour. The positive effect of *T. harzianum* treatment on the wheat primary root was obvious as its length increased by 2.8–3.8 cm depending on the cultivar. In addition to the direct effect on seed germination, some species of *Trichoderma* have a biocontrol effect and are used as biopesticides against fungal plant diseases [61]. The increase in soya bean seed germination by up to 20% was established after the treatment with six different species of *Trichoderma* had been applied [62].

#### 2.3.3. Seedling Infection and Growth

The seed treatment with *T. harzianum* showed the highest efficacy in reducing seedling infection with the pathogens of the genera *Fusarium*, *Alternaria*, *Penicillium,* and *Mucor*. Thus, in all tested cultivars, percentages of infected seedlings with *Fusarium* sp. and *Alternaria* sp. was 1% compared to the control that amounted to 4% and 3–4%, respectively, while the presence of *Penicillium* sp. and *Mucor* sp. was not detected in the treated seedlings (Table 4). The highest increase in the seedling length occurred when seeds were treated with *T. harzianum* and *Bacillus* isolates B5, B7, and B8 (Table 10).

#### 2.3.4. Correlation, PCA, and Cluster Analysis

In all wheat cultivars a negative but not significant (*p* ≥ 0.05) correlation between germination and the presence of examined pathogens was found (Table 11). Only in the cultivar Zemunska rosa, the correlation between germination and the presence of *Fusarium* sp. was significant (*p* ≤ 0.05) and negative (r = −0.649). Furthermore, a negative, but not significant correlation (*p* ≥ 0.05) was detected between the seedling growth (which includes the growth of primary roots and shoots), and the presence of the examined pathogens in all wheat cultivars. Contrarily, the highest (*p* ≤ 0.001) and positive dependence was established between seed germination and the primary shoot length in all tested cultivars (r = 0.911–0.983). The same applies to the correlation between germination and the primary root length in cultivars Salazar (r = 0.960) and Zemunska rosa (r = 0.903), while a correlation of a lower significance was obtained in the cultivar Apicol (r = 0.785). A positive correlation at the highest significance level (*p* ≤ 0.001) was determined between the shoot length and the root length in the cultivars Apicol (r = 0.893) and Salazar (r = 0.955), while in the cultivar Zemunska rosa it was at the lower significance level (r = 0.863) (Table 11).

Based on the examined effects of the treatments applied to three wheat cultivars and their effects on germination, presence of fungal pathogens and the shoot and the root length, the cluster analysis clearly singed out the impact of *T. harzianum* on all three examined cultivars. Treatments with *Bacillus* isolates (Figure 3 and Figure 4) affected cultivars Zemunska rosa (B5, B7 and B8) and Salazar (B8). The PCA confirmed that treatments with *T. harzianum* and *Bacillus* isolates B8, B5, and B7 were classified in the same group—A1 (Figure 3 and Figure 4). Treatments with B3 and B4 applied in to the cultivars Apicol and Zemunska rosa were also classified in the same group but closer to the border with the group B1.

## 3. Materials and Methods

### 3.1. Plant Material

The following cultivars (factor A) of lettuce (*Lactuca sativa* L.): (i) Genius, (ii) Majska kraljica, (iii) Endivia, celeriac (*Apium graveolens* L.): (i) Cezar, (ii) Omega, (iii) Praški krupni and wheat (*Triticum aestivum* L.): (i) Apicol, (ii) Salazar and (iii) Zemunska rosa were used in the study.

Studies with lettuce and celeriac were performed in two locations (factor C): the Seed Testing Laboratory in Topčider (with the germination cabinet and a greenhouse) (C1) and the Seed Testing Laboratory in Batajnica (with the germination cabinet and a greenhouse) (C2). Studies with wheat were performed in the Laboratory for Testing the Quality of Seeds and Planting Material of the Institute for Plant Protection and the Environment with germination cabinet during the trial was carried out according to the same procedure for two years, 2018–2019 (factor D).

The following parameters were observed in the study: seed germination, the shoot, and root length.

### 3.2. Preparation of Plant Material to Germination

In the germination cabinet, lettuce, celeriac, and wheat seeds were sown in boxes (size: 20 cm (L), 14 cm (W), and 4 cm (D), with sand as a growing medium with the grain size of 0.05–0.8 mm, pH 7, previously sterilized at 100 °C for 60 min. The sand moisture was maintained at the optimum level by occasional wetting. Seeds (4 × 100) were sown for each cultivar within each plant species.

In the greenhouse, lettuce and celeriac seeds were sown in 25 cm soil-filled pots. Four hundred seeds (4 × 100) were sown for each cultivar within each plant species.

### 3.3. Laboratory Condition of Germination

Germination cabinets: germination of lettuce seeds was tested at 20 °C, in the dark, for seven days. Germination of celeriac seeds was tested within the temperature regime 20 <=> 30 °C (altering temperature regime of 20 °C for 16 h in the dark and 30 °C for 8 h in the light) for 21 days. Germination of wheat in germination cabinets was tested in the following way: cold test—5–10 °C for three days and 20 °C for eight days in the dark.

In the greenhouse, germination was determined 10 and 30 days after sowing of lettuce and celeriac, respectively.

### 3.4. Stress Factors

There were four treatments (factor B) applied to lettuce and celeriac cultivars and compared to the control (without any treatments: 1) zinc (ZnSO_4_ 0.5%), 2) boron (0.025%), 3) biological product − Coveron (commercial product consisting of *Trichoderma atroviride*, *Glomus mosseae*, *Glomus intraradices)*, and 4) MIX − (zinc ZnSO_4_ + boron 0.025% + biological product − Coveron).

After the 2019 and 2020 wheat harvest, seeds of three wheat cultivars: Apicol, Salazar, and Zemunska rosa were separately treated with the isolate of *Trichoderma harzianum* and eight isolates of *Bacillus* spp. (from B1 to B8). Tested seed were treated with a spore suspension of *T. harzianum* at a concentration of 3.6 × 10^7^ CFU mL^−1^. *Bacillus* spp. isolates were grown on the Luria-Bertani (LB) medium and incubated at the temperature of 30 °C for 24 h and adjusted to a concentration of approximately 10^8^ CFU mL^−1^ [63]. Wheat seeds were soaked in isolates suspensions and kept for a 24 h, after which they were placed on germination according to the ISTA Rules [64] in the Laboratory for Testing the Quality of Seeds and Planting Material of the Institute for Plant Protection and the Environment. The seed control samples were treated with sterile water. Experiments were conducted in four replicates, each containing 100 seeds (4 × 100). After eight days of the germination test, the germination and the presence of fungal pathogens were evaluated. The presence of the following fungal genera commonly found in non-treated wheat seeds: *Fusarium* sp., *Alternaria* sp., *Penicillium* sp., and *Mucor* sp. were expressed as the percentage of infected seeds. Plant pathogens were identified based on standard agar plate methods used for detection of seed-borne fungi and morphology observed by light microscopy [64] (ISTA, 2020). The final seedling count was made after 8 days for all three cultivars. Primary root length (cm) and the primary shoot length (cm) were measured on healthy seedlings after the final counting. The seedling length was measured using a ruler.

The fungus *T. harzianum* was isolated in the Mycology Laboratory, Department of Plant Physiology of the Institute for Biological Research Siniša Stanković, Belgrade. Eight bacterial isolates (coded as B1–B8) belonging to the genus Bacillus were isolated from the soil in the Department of Plant Diseases of the Institute for Plant Protection and Environment, Belgrade.

### 3.5. Determination of Shoot and Root Length in Laboratory Conditions and Greenhouses

The shoot and root lengths in germination cabinets were determined on the same day as germination—on the 7th and 21st day for lettuce and celeriac, respectively, and 8th day for wheat, according to the method previously applied in the study carried out by Stanisavljević et al. [65]. In the greenhouses, the length of shoots and roots was determined on the 10th for lettuce and the 30th day for celery, after sowing on the pots, according to the method previously applied in the study by Stanisavljević et al. [65].

### 3.6. Statistical Analysis

The ANOVA (F test) was applied to determine the effect of factors. The Tukey’s multiple range test (*p* ≤ 0.05) and the coefficient of variance (CV%) were used to test environmental effects. Standard error of the mean was calculated to indicate variation around the mean (±s.e.m.). The Pearson’s correlation between germination and other examined parameters was calculated using simple correlation coefficients (r). Arcsine transformation was applied to seed germination data and percentages of infected seed before being subjected to the analysis of variance [66]. Cluster and principal component analysis (PCA) was performed based on comparing the impact of treatments. Acquired experimental data were processed using the freeware software package, R-Statistics [67] and Minitab [68]. Statistical analysis comprehended the three general phases.

## 4. Conclusions

The effect on germination and the initial growth of lettuce and celeriac seedlings treated with Zn, B, the biological preparation—Coveron, or with the mixture of these three treatments was significantly positive or showed a positive trend compared to the control. The application of the biological preparation Coveron can significantly improve germination and the initial growth of lettuce and celeriac seedlings in organic production. On the other hand, in conventional production of these two species, the great improvement can be achieved by the application of both Coveron and microelements or their mixture with Coveron.

The most pronounced effect of Coveron was on the root growth when seeds of both plant species were grown, after the treatment, in the greenhouse. This especially points out to positive effects after the product was used by farmers.

The significant cultivar × treatment interaction indicates the possibility of selecting the treatment adequate to each cultivar and achieving an additional effect on improving germination and the seedling growth of both the studied species.

Wheat seed treatments with *T. harzianum* and isolates of the genus *Bacillus* can increase germination of wheat seeds by 4% to 8%, respectively. The maximum germination was 91% in the cultivar Salazar treated with *T. harzianum* while the minimum was 83% in the cultivar Apicol treated with the isolate B1. The relatively high germination in relation to the control confirmed the positive impact of the applied treatments on this seed trait. Seed treatments with *T. harzianum* and selected *Bacillus* isolates caused a reduction of plant pathogenic fungi: *Fusarium* sp., *Alternaria* sp., *Penicillium* sp., and *Mucor* sp., by which a negative correlation between seed germination and the presence of pathogens in all examined wheat cultivars was confirmed. Since plant pests can significantly affect agricultural production, sustainable strategies for disease management are necessary in order to reduce the use of pesticides in agriculture. Due to the positive effects on the seed quality and the seedling growth, selected isolates of *Bacillus* sp. (B5, B7, B8) and especially fungus *T. harzianum* could be observed as biocontrol agents for control of plant pathogens in the organic system of wheat cultivation. Wheat cultivar × treatment interaction significantly contributed to improving germination and the seedling growth.

## Figures and Tables

**Figure 1 plants-10-01913-f001:**
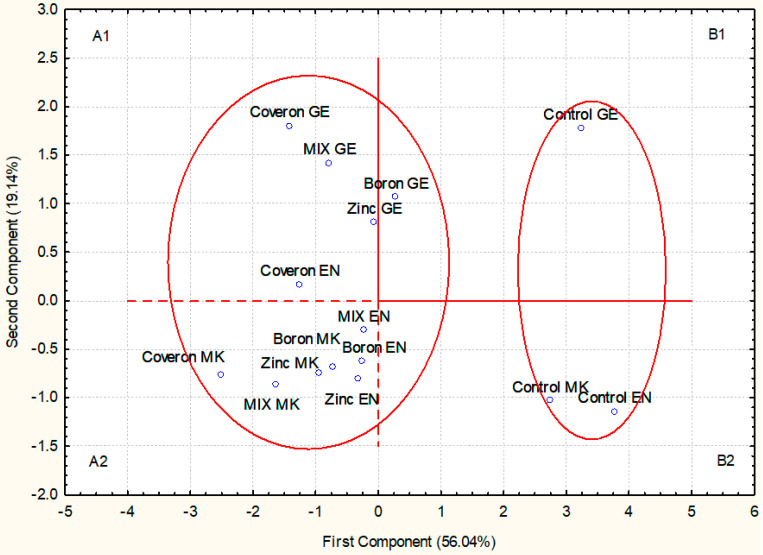
Principal component analysis (PCA) for the treatment effect on seed germination and shoot and root length of lettuce (*Lactuca sativa,* GE—Genius, MK—Majska kraljica, EN—Endivia.

**Figure 2 plants-10-01913-f002:**
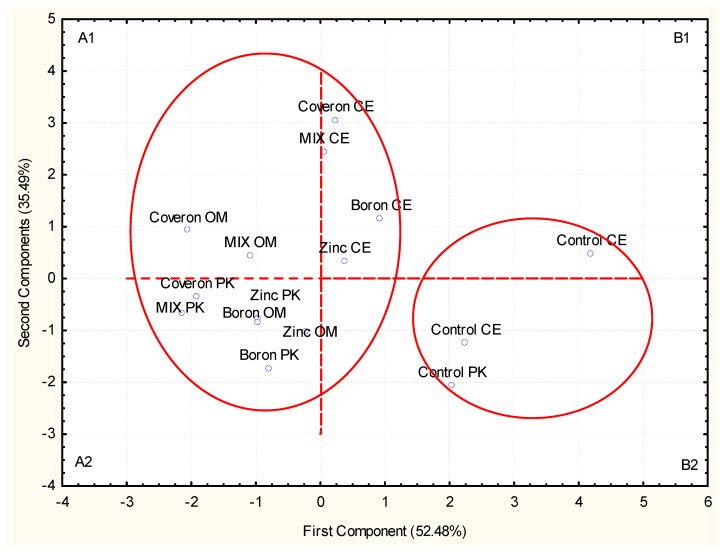
Principal component analysis (PCA) for the treatment effect on germination seed and on embryonic stem and radicle (*Apium graveolens* L.), C—Cezar, OM—Omega, PK—Praški krupni.

**Figure 3 plants-10-01913-f003:**
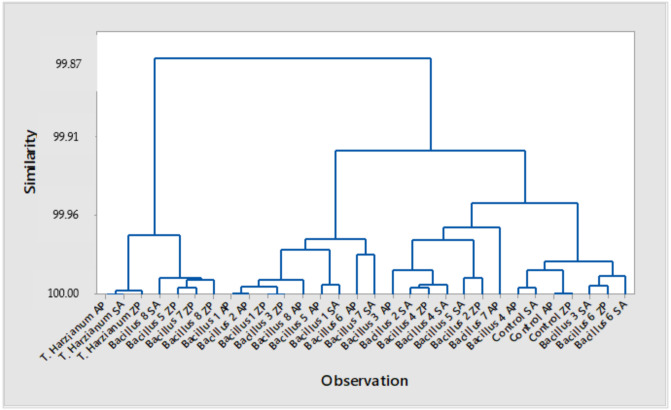
Cluster analysis for the treatment effect on germination, presence of fungal species *Fusarium* sp., *Alternaria* sp., *Penicillium* sp., *Mucor* sp., and on the shoot and root length, AP—Apicol, SA—Salazar, ZP—Zemunska rosa.

**Figure 4 plants-10-01913-f004:**
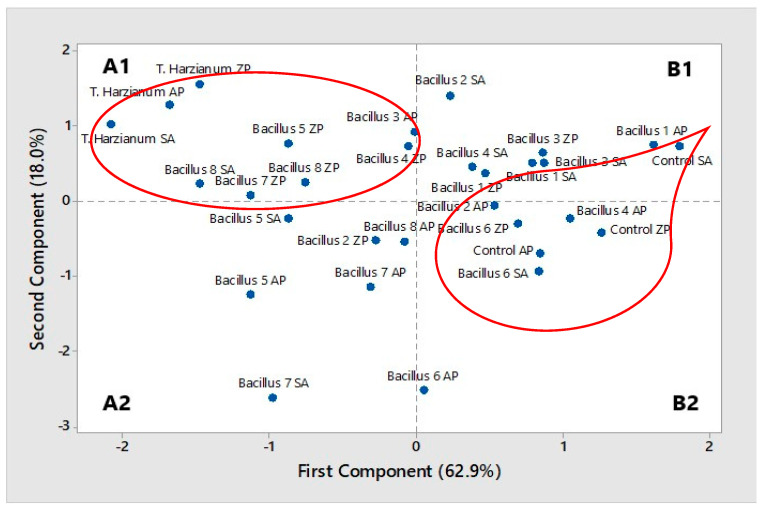
Principal component analysis (PCA) for the treatment effect on germination presence of fungi (*Fusarium* sp., *Alternaria* sp., *Penicillium* sp., *Mucor* sp.) and the shoot and root length, AP—Apicol, SA—Salazar, ZP—*Zemunska* rosa.

**Table 1 plants-10-01913-t001:** Germination after treatments applied to lettuce cultivars.

Results Obtained in Germination Cabinets (a)						
Traits	Treatment	Cultivar			Average	CV (%)
		Genius	Majska Kraljica	Endivia		
Germination						
(%)	Control	88 ± 0.58 b	86 ± 0.19 b	86 ± 0.12 b	87	1.33
	Zinc	89 ± 0.33 ab	89 ± 0.33 ab	91 ± 0.18 a	90	1.29
	Boron	89 ± 0.28 ab	88 ± 0.48 ab	91 ± 0.42 a	89	1.71
	MIX	91 ± 0.72 a	90 ± 0.63 a	90 ± 0.60 a	90	0.64
	Coveron	92 ± 0.52 a	91 ± 0.51 a	89 ± 0.50 ab	91	1.68
	Average	89	88	90	-	-
	CV %	1.41	1.94	2.66	-	-
**Results Obtained in Greenhouses (b)**						
	Control	74 ± 0.54 b	79 ± 0.12 c	83 ± 0.58 b	79	5.73
	Zinc	87 ± 0.63 a	95 ± 0.33 a	89 ± 0.58 a	90	4.61
	Boron	88 ± 0.81 a	94 ± 0.58 a	90 ± 0.58 a	91	3.37
	MIX	90 ± 0.45 a	93 ± 0.50 ab	88 ± 0.58 a	90	2.79
	Coveron	88 ± 0.16 a	91 ± 0.76 b	89 ± 0.58 a	89	1.71
	Average	85	90	85	-	-
	CV %	7.57	7.24	4.53	-	-

Tukey’s multiple range test, significant effect *p* ≤ 0.05, different small letters, a, b… x, values are mean ± standard error of the mean, Coveron − (commercial product consisting of *Trichoderma atroviride*, *Glomus mosseae*, *Glomus intraradices*), MIX − (zinc ZnSO_4_ + boron 0.025% + biological product − Coveron).

**Table 2 plants-10-01913-t002:** Growth of shoot length after treatments applied to lettuce cultivars.

Results Obtained in Germination Cabinets (a)						
Traits	Treatment	Cultivar			Average	CV (%)
		Genius	Majska Kraljica	Endivia		
Shoot length						
(cm)	Control	4.5 ± 0.16 b	3.7 ± 0.11 b	3.1 ± 0.13 b	3.8	18.6
	Zinc	4.6 ± 0.28 ab	4.2 ± 0.55 a	3.7 ± 0.16 a	4.2	10.8
	Boron	4.7 ± 0.58 ab	4.3 ± 0.68 a	3.8 ± 0.72 a	4.3	10.6
	MIX	4.8 ± 0.62 a	4.1 ± 0.19 a	3.6 ± 0.55 ab	4.2	14.5
	Coveron	5.0 ± 0.50 a	4.2 ± 0.38 a	4.3 ± 0.14 a	4.5	9.68
	Average	4.7	4.1	3.7	-	-
	CV %	4.08	5.72	11.6	-	-
**Results Obtained in Greenhouses (b)**						
	Control	5.9 ± 0.76 b	5.5 ± 0.14 b	5.6 ± 0.50 c	5.7	3.67
	Zinc	7.2 ± 0.32 a	6.9 ± 0.59 a	6.6 ± 0.14 b	6.9	4.35
	Boron	6.9 ± 0.12 a	7.0 ± 0.12 a	6.5 ± 0.16 b	6.8	3.89
	MIX	7.3 ± 0.45 a	7.3 ± 0.14 a	7.8 ± 0.25 a	7.5	3.87
	Coveron	7.6 ± 0.65 a	7.7 ± 0.62 a	8.1 ± 0.36 a	7.8	3.39
	Average	7.0	6.9	6.9	-	-
	CV %	9.36	12.1	14.8	-	-

Tukey’s multiple range test, significant effect *p* ≤ 0.05, different small letters, a, b… x, values are mean ± standard error of the mean Coveron − (commercial product consisting of *Trichoderma atroviride*, *Glomus mosseae*, *Glomus intraradices*), MIX − (zinc ZnSO_4_ + boron 0.025% + biological product − Coveron).

**Table 3 plants-10-01913-t003:** Growth of root length after treatments applied to lettuce cultivars.

Results Obtained in Germination Cabinets (a)						
Traits	Treatment	Cultivar			Average	CV (%)
		Genius	Majska Kraljica	Endivia		
Root length						
(cm)	Control	1.1 ± 0.11 b	1.8 ± 0.72 b	0.6 ± 0.42 b	1.2	51.7
	Zinc	1.2 ± 0.56 ab	2.1 ± 0.58 ab	1.3 ± 0.55 a	1.5	32.2
	Boron	1.3 ± 0.12 ab	2.1 ± 0.14 ab	1.0 ± 0.18 ab	1.5	38.8
	MIX	1.5 ± 0.62 a	2.2 ± 0.33 a	1.1 ± 0.14 ab	1.6	34.8
	Coveron	1.4 ± 0.50 a	2.4 ± 0.51 a	1.4 ± 0.56 a	1.7	33.3
	Average	1.3	2.1	1.1	-	-
	CV %	12.2	10.2	28.8	-	-
**Results Obtained in Greenhouses (b)**						
	Control	0.4 ± 0.48 b	0.6 ± 0.50 b	0.5 ± 0.77 b	0.5	20.0
	Zinc	0.7 ± 0.58 a	0.7 ± 0.66 ab	0.8 ± 0.65 a	0.7	7.87
	Boron	0.6 ± 0.52 a	0.7 ± 0.17 ab	0.8 ± 0.14 a	0.7	14.3
	MIX	0.6 ± 0.13 a	0.8 ± 0.53 a	0.7 ± 0.18 a	0.7	14.3
	Coveron	0.7 ± 0.55 a	0.9 ± 0.88 a	0.7 ± 0.22 a	0.8	15.1
	Average	0.6	0.7	0.7	-	-
	CV %	20.4	15.4	17.7	-	-

Tukey’s multiple range test, significant effect *p* ≤ 0.05, different small letters, a, b… x, values are mean ± standard error of the mean, Coveron − (commercial product consisting of *Trichoderma atroviride*, *Glomus mosseae*, *Glomus intraradices*), MIX − (zinc ZnSO_4_ + boron 0.025% + biological product − Coveron).

**Table 4 plants-10-01913-t004:** Factors on germination and the seedling growth after treatments applied to lettuce cultivars.

Results Obtained in Germination Cabinets (F-Test)			
Source	Genius	Majska Kraljica	Endivia
Cultivar (A)	**	*	*
Treatment (B)	**	*	*
Location (C)	ns	ns	ns
Year (D)	ns	ns	ns
A × B	*	*	*
A × C	ns	ns	ns
A × D	ns	ns	ns
B × C	ns	ns	ns
B × D	ns	ns	ns
C × D	ns	ns	ns
**Results Obtained in Greenhouses (F-Test)**			
Cultivar (A)	*	*	*
Treatment (B)	**	*	*
Location (C)	ns	ns	ns
Year (D)	ns	ns	ns
A × B	*	*	*
A × C	ns	ns	ns
A × D	ns	ns	ns
B × C	ns	ns	ns
B × D	ns	ns	ns
C × D	ns	ns	ns

Significancy of the F tests at the: * *p* ≤ 0.05, ** *p* ≤ 0.01; ns—not significant (*p* ≥ 0.05); interaction of factors: cultivar × treatment (A × B), cultivar × location (A × C), cultivar × year (A × D), treatment × location (B × C), teratment × year (B × D), location × year (C × D).

**Table 5 plants-10-01913-t005:** Germination after treatments applied to celeriac cultivars.

Results Obtained in Germination Cabinets (a)						
Traits	Treatment	Cultivar			Average	CV (%)
		Cezar	Omega	Praški Krupni		
Germination						
(%)	Control	64 ± 0.13 c	79 ± 0.54 b	80 ± 0.50 b	74	12.1
	Zinc	87 ± 0.28 a	86 ± 0.28 a	92 ± 0.45 a	88	3.64
	Boron	79 ± 0.58 ab	89 ± 0.34 a	95 ± 0.84 a	88	9.22
	MIX	82 ± 0.92 a	86 ± 0.22 a	95 ± 0.67 a	88	7.60
	Coveron	78 ± 0.19 b	87 ± 0.77 a	95 ± 0.55 a	87	9.81
	Average	78	85	91 a	-	-
	CV %	11.0	4.43	7.12	-	-
**Results Obtained in Greenhouses (b)**						
	Control	57 ± 0.19 c	69 ± 0.10 b	79 ± 0.41 c	68	16.1
	Zinc	82 ± 0.11 a	81 ± 0.18 a	87 ± 0.63 b	83	3.86
	Boron	72 ± 0.71 b	84 ± 0.84 a	91 ± 0.17 a	82	11.7
	MIX	72 ± 0.63 b	79 ± 0.52 ab	86 ± 0.63 b	79	8.86
	Coveron	69 ± 0.54 b	82 ± 0.32 a	90 ± 0.48 a	80	13.2
	Average	70	79	87	-	-
	CV %	12.7	7.44	5.45	-	-

Tukey’s multiple range test, significant effect *p* ≤ 0.05, different small letters, a, b… x, values are mean ± standard error of the mean, Coveron − (commercial product consisting of *Trichoderma atroviride*, Glomus *mosseae*, *Glomus intraradices*), MIX − (zinc ZnSO_4_ + boron 0.025% + biological product − Coveron).

**Table 6 plants-10-01913-t006:** Growth of shoot length after treatments applied to celeriac cultivars.

Results Obtained in Germination Cabinets (a)						
Traits	Treatment	Cultivar			Average	CV (%)
		Cezar	Omega	Praški krupni		
Shoot length						
(cm)	Control	1.0 ± 0.63 b	1.8 ± 0.55 b	1.8 ± 0.44 c	1.5	30.1
	Zinc	1.2 ± 0.18 a	2.5 ± 0.63 a	2.3 ± 0.56 b	2.0	35.0
	Boron	1.3 ± 0.77 a	2.4 ± 0.98 a	2.2 ± 0.25 b	2.0	29.8
	MIX	1.3 ± 0.28 a	2.3 ± 0.13 a	2.7 ± 0.15 a	2.1	34.3
	Coveron	1.2 ± 0.14 a	2.6 ± 0.42 a	2.8 ± 0.76 a	2.2	39.6
	Average	1.2	2.3	2.4	-	-
	CV %	10.2	13.4	17.1	-	-
**Results Obtained in Greenhouses (b)**						
	Control	1.7 ± 0.67 c	1.3 ± 0.09 c	1.2 ± 0.72 b	1.4	18.9
	Zinc	2.4 ± 0.53 b	1.8 ± 0.17 b	2.0 ± 0.67 a	2.1	14.8
	Boron	2.3 ± 0.45 b	1.8 ± 0.54 b	1.8 ± 0.14 a	2.0	14.7
	MIX	3.0 ± 0.62 a	2.1 ± 0.82 ab	1.9 ± 0.17 a	2.3	25.1
	Coveron	3.1 ± 0.13 a	2.4 ± 0.43 a	2.0 ± 0.43 a	2.5	22.3
	Average	2.5	1.9	1.8	-	-
	CV %	22.8	21.7	18.8	-	-

Tukey’s multiple range test, significant effect *p* ≤ 0.05, different small letters, a, b… x, values are mean ± standard error of the mean, Coveron − (commercial product consisting of *Trichoderma atroviride*, *Glomus mosseae, Glomus intraradices*), MIX − (zinc ZnSO_4_ + boron 0.025% + biological product − Coveron).

**Table 7 plants-10-01913-t007:** Growth of root length after treatments applied to celeriac cultivars.

Results Obtained in Germination Cabinets (a)						
Traits	Treatment	Cultivar			Average	CV (%)
		Cezar	Omega	Praški krupni		
Root length						
(cm)	Control	1.2 ± 0.43 b	1.1 ± 0.63 b	1.0 ± 0.64 c	1.1	9.09
	Zinc	1.4 ± 0.16 ab	1.4 ± 0.78 ab	1.3 ± 0.53 b	1.4	4.22
	Boron	1.5 ± 0.75 a	1.6 ± 0.25 a	1.3 ± 0.17 b	1.5	10.4
	MIX	1.5 ± 0.19 a	1.7 ± 0.65 a	1.8 ± 0.13 a	1.7	9.17
	Coveron	1.6 ± 0.55 a	1.8 ± 0.13 a	1.7 ± 0.26 a	1.7	5.88
	Average	1.4	1.5	1.4	-	-
	CV %	10.5	18.3	23.0	-	-
**Results Obtained in Greenhouses (b)**						
	Control	2.6 ± 0.38 c	2.9 ± 0.19 c	2.7 ± 0.11 c	2.7	5.59
	Zinc	3.3 ± 0.56 b	3.3 ± 0.11 b	3.8 ± 0.55 a	3.5	8.33
	Boron	3.6 ± 0.13 ab	3.5 ± 0.77 b	3.2 ± 0.61 b	3.4	6.06
	MIX	4.3 ± 0.10 a	4.0 ± 0.39 ab	3.7 ± 0.13 a	4.0	7.50
	Coveron	4.4 ± 0.17 a	4.4 ± 0.46 a	3.9 ± 0.52 a	4.2	6.82
	Average	3.6	3.6	3.5	-	-
	CV %	20.4	16.3	14.5	-	-

Tukey’s multiple range test, significant effect *p* ≤ 0.05, different small letters, a, b… x, values are mean ± standard error of the mean, Coveron − (commercial product consisting of *Trichoderma atroviride*, *Glomus mosseae, Glomus intraradices*), MIX − (zinc ZnSO_4_ + boron 0.025% + biological product − Coveron).

**Table 8 plants-10-01913-t008:** Factors on germination and the seedling growth after treatments applied to celeriac cultivars.

Results Obtained in Germination Cabinets (F-Test)			
Source	Cezar	Omega	Praški krupni
Cultivar (A)	*	*	*
Treatment (B)	**	*	*
Location (C)	ns	ns	ns
Year (D)	ns	ns	ns
A × B	*	*	*
A × C	ns	ns	ns
A × D	ns	ns	ns
B × C	ns	ns	ns
B × D	ns	ns	ns
C × D	ns	ns	ns
**Results Obtained in Greenhouses (F-Test)**			
Cultivar (A)	*	*	*
Treatment (B)	**	*	*
Location (C)	ns	ns	ns
Year (D)	ns	ns	ns
A × B	*	*	*
A × C	ns	ns	ns
A × D	ns	ns	ns
B × C	ns	ns	ns
B × D	ns	ns	ns
C × D	ns	ns	ns

Significancy of the F tests at the: * *p* ≤ 0.05, ** *p* ≤ 0.01; ns—not significant (*p* ≥ 0.05); interaction of factors: cultivar × treatment (A × B), cultivar × location (A × C), cultivar × year (A × D), treatment × location (B × C), teratment × year (B × D), location × year (C × D).

**Table 9 plants-10-01913-t009:** ANOVA for assess the influence of year, cultivar and treatment on the tested features: germination (%), *Fusarium* sp. (%), *Alternaria* sp. (%), *Penicillium* sp. (%), *Mucor* sp. (%), shoot, and root length (cm).

Factor	Feature						
	G	F	A	*P*	M	SL	RL
D	ns	ns	ns	ns	ns	ns	ns
A	*	**	**	**	**	*	*
B	***	***	***	***	***	*	*
D × B	ns	ns	ns	ns	ns	ns	ns
D × A	ns	ns	ns	ns	ns	ns	ns
A × B	*	*	*	**	**	*	*

Significancy of the F tests at the: * *p* ≤ 0.05, ** *p* ≤ 0.01, *** *p* ≤ 0.001; ns—not significant (*p* ≥ 0.05); G—germination (G); F—*Fusarium* sp. (%); A—*Alternaria* sp. (%); P—*Penicillium* sp. (%); M—*Mucor* sp. (%); SL—shoot length and RL—root length (cm). D—year, A—cultivar, B—treatment, Interaction of factor: year × treatment (D × B), year × cultivar (D × A), cultivar × treatment (A × B).

**Table 10 plants-10-01913-t010:** Influence of *Trichoderma harzianum* isolate and eight different isolates of *Bacillus* spp. to influence germination, seedling growth, and the presence of some plant-pathogenic fungi (*Fusarium* sp., *Alternaria* sp., *Penicillium* sp., and *Mucor* sp.), seed germination, and initial seedling growth (primary shoot and root length) on seeds of three wheat cultivars.

C	Treatment	F	A	P	M	G	SL	RL
Apicol	Control	4 ± 0.96 b	4 ± 0.00 a	1 ± 0.58 a	1 ±0.00 ab	86 ± 0.48 b	6.01 ± 0.31 b	10.6 ± 0.95 b
	*Trichoderma* *harzianum*	1 ± 0.58 c	1 ± 0.58 b	0 ± 0.00 b	0 ± 0.00 b	90 ± 0.29 a	7.56 ± 0.33 a	13.4 ± 0.66 a
	*Bacillus* 1	4 ± 0.96 b	3 ±0.58 ab	0 ± 0.50 b	0 ± 0.50 b	83 ± 0.31 b	6.03 ± 0.81 b	11.1 ± 0.75 ab
	*Bacillus* 2	4 ± 0.82 b	3 ±0.82 ab	0 ± 0.00 b	0 ± 0.00 b	86 ± 0.19 b	6.52 ± 0.48 ab	11.4 ± 0.89 ab
	*Bacillus* 3	2 ± 0.96 c	2 ± 0.96 b	1 ± 0.00 a	1 ± 0.58 ab	87 ± 0.18 ab	6.67 ±0.51 ab	11.2 ± 0.88 ab
	*Bacillus* 4	4 ± 0.96 b	4 ± 0.82 a	0 ± 0.50 b	0 ± 0.00 b	85 ± 0.09 b	6.56 ± 0.67 ab	10.8 ± 0.81 b
	*Bacillus* 5	5 ± 0.78 ab	2 ± 0.96 b	0 ± 0.00 b	0 ± 0.00 b	90 ± 0.22 a	7.52 ± 0.51 a	12.8 ± 0.55 a
	*Bacillus* 6	6 ± 0.65 a	4 ± 0.50 a	0 ± 0.00 b	2 ± 0.96 a	88 ± 0.36 ab	6.99 ± 0.80 ab	11.9 ± 0.46 ab
	*Bacillus* 7	3 ± 0.82 bc	4 ± 0.50 a	0 ± 0.50 b	3 ± 0.58 a	88 ± 0.21 ab	7.03 ± 0.71 a	12.6 ± 0.81 a
	*Bacillus* 8	4 ± 0.00 b	3 ± 0.58 ab	1 ± 0.58 a	1 ± 0.58 ab	87 ± 0.36 ab	6.98 ± 0.55 a	12.9 ± 0.92 a
Salazar	Control	4 ± 0.96 b	3 ± 0.00 ab	0 ± 0.00 b	0 ± 0.00 b	83 ± 0.26 c	6.06 ± 0.52 b	10.1 ± 0.86 b
	*Trichoderma* *harziaum*	1 ± 0.58 c	1 ± 0.00 b	0 ± 0.00 b	0 ± 0.00 b	91 ± 0.26 a	7.77 ± 0.44 a	13.7 ± 0.38 a
	*Bacillus* 1	5 ± 0.29 ab	1 ± 0.58 b	0 ± 0.00 b	0 ± 0.00 b	85 ± 0.38 bc	6.65 ± 0.55 ab	11.2 ± 1.08 ab
	*Bacillus* 2	2 ± 0.00 c	2 ± 0.00 b	0 ± 0.00 b	0 ± 0.00 b	86 ± 0.22 b	6.72 ± 031 ab	11.5 ± 0.55 ab
	*Bacillus* 3	3 ± 0.82 bc	3 ± 0.58 ab	0 ± 0.50 b	1 ± 0.00 ab	85 ± 0.26 bc	6.69 ± 0.48 ab	10.9 ± 0.28 ab
	*Bacillus* 4	3 ± 0.82 bc	3 ± 0.58 ab	1 ± 0.00 a	0 ± 0.50 b	86 ± 0.38 b	6.71 ± 0.68 ab	11.7 ± 0.77 ab
	*Bacillus* 5	2 ± 0.96 c	4 ± 0.00 a	0 ± 0.00 b	0 ± 0.00 b	89 ± 0.17 ab	7.21 ± 0.78 a	12.9 ± 0.81 a
	*Bacillus* 6	4 ± 0.82 b	4 ± 0.96 a	0 ± 0.50 b	2 ± 0.00 a	86 ± 0.32 b	6.66 ± 0.66 ab	10.4 ± 0.89 b
	*Bacillus* 7	6 ± 0.46 a	4 ± 0.82 a	0 ± 0.50 b	0 ± 0.00 b	90 ± 0.24 a	7.71 ± 0.51 a	13.3 ± 0.98 a
	*Bacillus* 8	3 ± 0.00 bc	1 ± 0.58 b	1 ± 0.58 a	0 ± 0.00 b	90 ± 0.22 a	7.68 ± 0.63 a	13.2 ± 0.75 a
Zemunska Rosa	Control	4 ± 0.96 a	4 ± 0.96 a	1 ± 0.58 a	1 ± 0.00 ab	85 ± 0.32 c	5.59 ± 0.67 b	10.3 ± 0.62 b
	*Trichoderma* *harzianum*	1 ± 0.00 c	1 ± 0.00 b	0 ± 0.00 b	0 ± 0.00 b	89 ± 0.41 a	7.59 ± 0.28 a	14.1 ± 0.46 a
	*Bacillus* 1	4 ± 0.96 a	2 ± 0.00 b	0 ± 0.00 b	0 ± 0.00 b	86 ± 0.32 bc	6.35 ± 0.31 ab	11.3 ± 0.35 ab
	*Bacillus* 2	3 ± 0.82 ab	4 ± 0.50 a	0 ± 0.50 b	0 ± 0.00 b	88 ± 0.26 ab	7.55 ± 0.52 ab	11.7 ± 0.68 ab
	*Bacillus* 3	4 ± 0.00 a	2 ± 0.96 b	0 ± 0.00 b	0 ± 0.00 b	85 ± 0.31 c	6.08 ± 0.83 b	11.1 ± 0.70 ab
	*Bacillus* 4	2 ± 0.96 b	3 ± 0.58 ab	0 ± 0.00 b	0 ± 0.50 b	87 ± 0.36 b	6.62 ± 0.74 ab	11.8 ± 0.68 ab
	*Bacillus* 5	3 ± 0.82 ab	1 ± 0.58 b	0 ± 0.50 b	0 ± 0.50 b	88 ± 0.48 ab	7.58 ± 0.51 a	13.5 ± 0.87 a
	*Bacillus* 6	4 ± 0.96 a	3 ± 0.82 ab	0 ± 0.00 b	1 ± 0.58 ab	86 ± 0.31 bc	6.59 ± 0.43 ab	10.6 ± 0.53 b
	*Bacillus* 7	3 ± 0.00 ab	2 ± 0.96 b	0 ± 0.00 b	0 ± 0.00 b	89 ± 0.17 a	7.62 ± 0.49 a	13.6 ± 0.88 a
	*Bacillus* 8	4 ± 0.82 a	1 ± 0.00 b	0 ± 0.50 b	0 ± 0.00 b	88 ± 0.21 ab	7.53 ± 0.82 a	13.4 ± 0.81 a

C-cultivar; Tukey test—significance level *p* ≤ 0.05, different small letters, a, b… x, values are mean ± standard error of the mean; was used to assess the mean treatment, for each cultivar separately; F—*Fusarium* sp. (%); A—*Alternaria* sp. (%); P—*Penicillium* sp. (%); M—*Mucor* sp. (%); G—germination (%); SL—shoot length and RL—root length (cm).

**Table 11 plants-10-01913-t011:** Correlation dependence (r) between the germination and other examined parameters in three wheat cultivars (*n* = 10).

Cultivar	Examined Feature	F	A	P	M	PSL	PRL
Apicol	Germination	−0.181 ns	−0.488 ns	−0.106 ns	−0.199 ns	0.911 ***	0.785 **
	*Fusarium* sp.	-	0.595 ns	−0.178 ns	0.106 ns	−0.174 ns	−0.522 ns
	*Aternaria* sp.		-	0.000 ns	0.510 ns	−0.535 ns	−0.522 ns
	*Penicillium* sp.			-	0.134 ns	−0.301 ns	−0.211 ns
	*Mucor* sp.				-	−0.087 ns	−0.146 ns
	Primary shoot length					-	0.893 ***
Salazar	Germination	−0.257 ns	−0.150 ns	−0.177 ns	−0.264 ns	0.983 ***	0.960 ***
	*Fusarium* sp.	-	0.306 ns	−0.106 ns	0.121 ns	−0.165 ns	−0.268 ns
	*Aternaria* sp.		-	−0.250 ns	0.416 ns	−0.213 ns	−0.240 ns
	*Penicillium* sp.			-	−0.234 ns	0.191 ns	0.228 ns
	*Mucor* sp.				-	−0.271 ns	−0.505 ns
	Primary shoot length					-	0.955 ***
Zemunska rosa	Germination	−0.649 *	−0.459 ns	−0.484 ns	−0.553 ns	0.954 ***	0.903 ***
	*Fusarium* sp.	-	0.223 ns	0.272 ns	0.408 ns	−0.495 ns	−0.589 ns
	*Aternaria* sp.		-	0.515 ns	0.545 ns	−0.500 ns	−0.562 ns
	*Penicillium* sp.			-	0.667 *	−0.614 ns	−0.467 ns
	*Mucor* sp.				-	−0.572 ns	−0.643 *
	Primary shoot length					-	0.863 **

* *p* ≤ 0.05, ** *p* ≤ 0.01, *** *p* ≤ 0.001, ns—not significant (*p* ≥ 0.05); F—*Fusarium* sp.; A—*Alternaria* sp.; P—*Penicillium* sp.; M—*Mucor* sp.; PSL—primary shoot length; PRL—primary root length.

## Data Availability

All data are available in manuscript.

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
