# Peer review of "Germination and the Initial Seedling Growth of Lettuce, Celeriac and Wheat Cultivars after Micronutrient and a Biological Application Pre-Sowing Seed Treatment"

_plants, 2021, doi:10.3390/plants10091913_

Round 1

Reviewer 1 Report

The research paper fit perfectly within the scope of the Journal. The authors assessed the effect of a biological commercial product ‘Coveron’ containing beneficial microorganisms including Trichoderma on the germination rate and seedling performance in terms of shoot and root growth of several agricultural and horticultural species. I have several concerns before accepting the paper for publication:

The title is too long and confusing the authors should adopt a shorter and direct title.

The abstract should be re-written: the abstract should start with one sentence about the general introduction of the work, when what was done and 6-7 sentences about the results highlights. The differences between treatments should be report in percentage in order to be clearer for the readers of Plants MDPI.

I notice that the unique significant difference was recorded between the treatment x cultivar while the other two factors did not affect the variables measured why ? this should be explained by the authors in the discussion section.

I urge the authors to merge the results and discussion section in order to avoid redundancy. The discussion section is the weak point of this work, it is not enough to say that your results are in line or not with the literature review by why, in other words what is the mechanism behind these differences.

The paper could not be accepted in the current form but it will be considered for publication after major revisions.

Author Response

Dear Reviewer, 

thank you for reviewing manuscript.

Reviewer 2 Report

Dear Authors,

The topic discussed by the authors is interesting, but the paper requires correction. I included most of the comments in the text. However, I would like to emphasize that in the introduction the authors should broaden the information on the stress factors and plant species studied. The results should be more compiled and divided into subsections depending on the parameters tested. This applies to both text and tables and figures. The discussion is largely devoted to wheat, in my opinion there is too little information about the other two species. In addition, the discussion is a part of the discussion of the results in relation to the literature data. I suggest cite there not only tables, but also figures. The material and methods in this form are illegible, require sub-chapters and a detailed description of the conducted experiments.

In terms of editing, the work requires thorough corrections in accordance with the requirements of the journal. There are many mistakes in it - I have selectively marked some of them. Besides, in my opinion, it requires a linguistic correction.

References are not prepared according to the requirements - I have marked examples of remarks directly in the text, but the whole thing should be checked and corrected. 

Author Response

Dear Reviewer,

thank you for reviewing the manuscript.

Round 2

Reviewer 1 Report

I am pleased to inform you that the revised manuscript has been significantly improved I have no more comments thus the paper is accepted for publication in Plants MDPI.

Congrats!

Author Response

Dear rewiever,

thank you for accepting the corrections to the manuscript.

Best regards,

Authors

Reviewer 2 Report

Dear Authors,

Minor changes have been made to the work, but there are still gaps and mistakes. Part marked in the text. The work still requires an editorial correction in its entirety. Check text for paragraphs, spaces, italics, and other journal requirements. Please review the references again. Figures are illegible (out of focus), please correct it. Delete unnecessary descriptions in the tables.

Author Response

Dear rewiever,

thank you to the rewiev.

Best regards,

Authors
